# Research Overview and Trends of the Effects of Gibberellins (GAs) on Rice Biological Processes: A Bibliometric Analysis

**DOI:** 10.3390/plants13111548

**Published:** 2024-06-03

**Authors:** Yifan Shen, Lijia Li

**Affiliations:** State Key Laboratory of Hybrid Rice, College of Life Sciences, Wuhan University, Wuhan 430072, China; yifanshen@whu.edu.cn

**Keywords:** gibberellins, rice, academic visualization, bibliometrics

## Abstract

Rice (*Oryza sativa* L.) is a vital crop that feeds more than half of the world’s population. Gibberellins (GAs), a crucial phytohormone, play a significant role in the growth and development of rice. Since 1985, there has been a notable increase in the number of studies investigating the effects of GA on various biological processes in rice. Nevertheless, conducting scientific and quantitative research on the extensive literature available poses significant challenges, particularly in understanding the development trajectory of the field, examining major contributors, and identifying emerging research trends. The objective of this study is to address these challenges by analyzing global research patterns and trends using bibliometric methods from 1985 to 2024. Through the application of advanced analytical tools, progress in this field is studied in depth and the global research landscape is characterized from multiple dimensions including countries, institutions, authors, and journals. The analysis of 2118 articles extracted and screened from the Web of Science Core dataset shows a steady growth in the number of publications. The research published in China and the USA has significantly advanced the development of the field. In particular, institutions such as the Chinese Academy of Sciences and Nagoya University have shown impressive productivity. Lee In-Jung stands out as the most influential author. The journal Plant Physiology publishes the highest number of articles. The study also provides a thorough examination of current research hotspots, indicating a predominant focus on understanding the role of GAs in the biological processes that regulate diverse rice phenotypes, including plant height, seed dormancy, germination, and stress resistance. By tracing the development characteristics and key points in this area, this study contributes to a quantitative and comprehensive understanding of the impact of GAs on rice.

## 1. Introduction

Rice is a staple food that feeds more than half of the global population, and China in particular, as one of the largest producers and consumers of rice in the world, plays a crucial role in ensuring world food security [1]. Rice growth and development are regulated by various hormones and environmental factors. Among these factors, gibberellins (GAs) are particularly important. GAs are a group of plant hormones that are cyclic diterpenoid compounds, meaning that they are derived from the isoprenoid biosynthetic pathway. These hormones were first discovered in the pathogenic fungus *Gibberella fujikuroi*, which induces the excessive elongation of the infected rice [2]. The significance of GAs in agriculture is highlighted by their role in the green revolution of the 1960s. One of the key developments during this time was the creation of dwarf crop varieties. These dwarf crops were the result of defects in GA synthesis and metabolism, which led to a reduction in plant height without compromising yield [3]. Although numerous forms of GAs exist in organisms, only a fraction are biologically active, with the rest serving as the precursors or inactive metabolites of active compounds. Key bioactive GAs include GA1, GA3, GA4, and GA7 [4]. The metabolism of GAs can be influenced by external environmental factors, and their functions are integral throughout the entire life cycle of a plant. This makes GAs a critical component of the complex signaling network that plants use to respond to their environment. Over the past twenty years, genetic screenings in rice and Arabidopsis have helped to identify the components of the GA signaling pathway. In addition to promoting overall plant growth through cell expansion and division, GAs are also involved in regulating a number of developmental processes throughout the plant’s life cycle. These processes include seed germination, photomorphogenesis, the transition to flowering, male fertility, and fruit development [5].

Transcriptional regulation controlled by GAs involves the degradation of transcriptional regulators triggered by the interaction of active GA molecules with receptors [6]. Active GA molecules bind to a receptor known as GA-insensitive DWARF1 (GID1), which triggers a conformational change that allows GID1 to interact with DELLA proteins, forming a GA-GID1-DELLA complex. The formation of the complex leads to the polyubiquitination and subsequent degradation of DELLA proteins by the proteasome. DELLA proteins are transcriptional regulators that interact with various transcription factors to control gene expression [7]. DELLA proteins act as inhibitors within the GA signaling network, influencing many processes regulated by GAs. Nonetheless, recent research has shown that not all GA signaling pathways are dependent on DELLA proteins, indicating a more complex regulatory system. The study of GAs’ effects on rice has been ongoing since 1985, with many research publications resulting from this work. Identifying key research areas and understanding the trends in the field of GA research presents a challenge. Since the green revolution, the significance of GAs in influencing rice phenotypes has been well established. There is a particular interest in how to use the knowledge of GAs’ influence on rice to guide agricultural production. To provide a comprehensive understanding and to inform future research directions, a thorough quantitative analysis is necessary to organize and highlight current research.

Traditional reviews summarize the progress and advancements in a specific research area, which are limited by the author’s subjective perspective, meaning they may not provide an objective or comprehensive view, while research articles focus on specific issues or questions within a research field, which offer in-depth analysis on particular topics, but they might not cover the broader scope of the field, thus lacking a holistic overview [8]. Bibliometric analysis is a quantitative and objective method of analyzing research literature. It uses a large number of papers to provide a comprehensive and unbiased view of the research landscape [9]. Tools for bibliometric analysis like VOSviewer and CiteSpace are mentioned as tools that can visually represent information within specific fields. These tools can help in understanding the research domain from various angles through visual representations [10]. By analyzing core journals, publications, and authors through co-citation and collaboration networks, researchers can gain insights into the historical development and current trends in the field. This can help in understanding the evolution of research and the interconnections between different areas of study. This study provides a clear overview of the publication trends related to the effects of GAs on rice biological processes, highlighting the growth and interest in this specific research area over the past several decades.

## 2. Results

### 2.1. Publication Outputs and Trends

Based on a literature search and screening strategy outlined in Figure 1, focusing on the publication trends within a specific research field related to the effects of gibberellins (GAs) on rice biological processes from 1 January 1985 to 29 February 2024, a total of 2118 studies were identified, which includes 1976 articles (93.3%) and 142 reviews (6.7%). This indicates a strong preference for original research articles over review articles in this field. The average annual publication rate is calculated to be 52.95, which suggests a consistent flow of new research being published each year. The cumulative number of publications has shown an upward trend over time, indicating growing interest and activity in the field. To analyze this trend further, an exponential function y = 1.7154x^2^ − 18.571x + 71.301 (R^2^ = 0.9959, where x represents the first year and y represents the cumulative number of articles) was used to model the annual publication pattern. This model demonstrated a well-fitted curve, as illustrated in Figure 2. The number of papers published in this field is steadily increasing each year, which can be attributed to factors such as the accessibility and knowledge sharing facilitated by open-access journals. The study of the impact of genetic algorithms on rice biological processes has become a prominent area of interest within the field. By considering various factors collectively, it is possible to anticipate future research and publication trends in this field.

### 2.2. Country Partnerships and Publications

This study on the impact of GAs on rice bioprocesses encompasses 69 countries/regions. To visualize national collaborative networks, a minimum threshold of four publications per country/region is applied in Figure 3A. China stands out with 968 publications, making up 45.70% of the total research output in this field. Following China, Japan and the United States of America contributed 20.73% (439 publications) and 14.02% (297 publications), respectively, indicating the global interest and engagement in the study of GAs’ effects on rice biological processes.

The peripheral curved segment indicates different countries and regions in the chord chart and its length indicates the corresponding published volume [11]. The degree of interconnection between countries represents their level of cooperation. During the study period, significant international collaborations were observed, with academic collaborations between China and the United States being the most frequent, demonstrating a significant linkage strength of 91 (relative value) (Figure 3B). The number of publications serves as an indicator of the prominence of a country or region in the field [12]. China, Japan, and the United States stand out as leaders in publications related to the effects of GAs on rice bioprocesses, showcasing their significant academic influence. This collaboration can facilitate the exchange of ideas, optimize resource utilization, and potentially expedite the advancement and enhancement of theories to effectively address common challenges. Such academic collaborations between different countries potentially help the exchange of ideas and technology, leading to more publications.

Citation bursts are essential for identifying publications that experience significant spikes in citations within a specific timeframe [11]. These bursts offer insights into dynamic trends and directions within the field of study, allowing for the analysis of highly cited publications to pinpoint emerging trends in scholarly interest. Figure 3C displays the citation bursts for the top 10 countries/regions, with red lines representing the magnitude of each burst. Notably, between 1988 and 2008, there was a notable surge in citations for publications from Japan (intensity = 90.38), closely followed by China (intensity = 47.47), indicating the emerging trends and areas of high scholarly interest. It is recommended that researchers and industry professionals focus on these countries and their research directions to gain valuable insights and potential collaboration opportunities. By leveraging this information, stakeholders can identify potential partners and monitor research collaborations, enhancing their understanding of the global distribution of expertise in the field.

### 2.3. Institutional Performance

Identifying the top-performing institutions and analyzing their citation bursts provide valuable insights for researchers seeking collaborators and funding opportunities. Dynamic collaborative networks among institutions signify a vibrant research ecosystem. Over the past 40 years, global research on the impact of GAs on biological processes in rice has made significant progress, involving more than 1480 entities. A cluster analysis of these inter-institutional collaborations, as depicted in Figure 4, required a minimum of nine publications. In the figure, each institution is represented by a circle, with the size of the circle indicating the number of publications. The thickness of the lines connecting the circles reflects the intensity of collaboration between institutions. The Chinese Academy of Sciences (Chinese Acad Sci) leads in the number of publications with 168 papers, followed by Nagoya Univ and Chinese Acad Agr Sci with 120 and 118 papers, respectively. Notably, Chinese Acad Sci and Univ Chinese Acad Sci exhibit the closest collaboration. It is interesting to observe that the majority of these organizations prioritize national collaborations over international ones. The active inter-institutional collaborations demonstrate a dynamic research ecosystem in the field of GAs’ impact on biological processes in rice, promoting innovation and knowledge exchange. This ecosystem serves as a fertile ground for sharing ideas, resources, and expertise, enabling researchers to stay updated on the latest developments, access shared resources, and enhance the quality of their research.

Identifying organizations experiencing citation bursts can assist researchers in recognizing active and influential research programs for potential collaboration or funding opportunities. When considering collaborations, it is crucial to evaluate not only the number of publications but also the enduring impact and adaptability of the research over time [13]. Through a CiteSpace analysis, this study pinpointed a citation surge at the National Institute of Agrobiological Sciences from 2002 to 2011, with a peak burst intensity of 27.46 (Figure 5A). RIKEN exhibited a citation surge from 1996 to 2007, particularly in the field of the effect of GAs on biological processes in rice, although this trend has waned in recent years. Conversely, since 2020, the University of Chinese Academy of Sciences, Chinese Academy of Agricultural Sciences, South China Agricultural University, and Yangzhou University have all experienced a notable increase in citations, signifying that their research findings have garnered recognition in recent years and have had a significant academic impact.

### 2.4. Author Contributions

Analyzing the network of prolific authors and their collaborations can aid researchers in identifying key contributors to the field. Mainstream co-authors can offer valuable insights and help shape future research directions. A comprehensive analysis of authorship in the realm of the effects of GAs on biological processes in rice revealed a total of 9762 authors who have collectively published 2118 related articles. To delve deeper into the co-authorship network, visual maps were generated using the VOSviewer software program, setting a minimum threshold of seven publications per author. The size of the circles in these maps corresponds to the number of papers authored by each individual, while different colors signify distinct clusters of authors. The thickness of connecting lines indicates the level of collaboration between authors. Notably, 80 authors surpassed the threshold, as depicted in Figure 6, with Lee, In-Jung, exhibiting the strongest collaboration alongside Khan, Abdul Latif, and Hamayun, Muhammad, each with 34 papers and a collaboration intensity of 33. Furthermore, Lee, In-Jung, was commended for his significant contribution to the research on the effects of GAs on rice biological processes, with a total of 63 publications, underscoring their noteworthy impact in the scientific domain.

Citation burst analysis is a crucial metric for measuring the frequency of citations within a specific research field [14]. The top 10 citation bursts for authors in the realm of the effects of GAs on biological processes in rice were examined by using CiteSpace for the period from 1 January 1985, to 29 February 2024 (Figure 5B). Matsuoka, M, held a distant second place with a burst score of 16.35 (2000–2006), while Lee, In-Jung, took the lead with a score of 15.08 in the period of 2009–2019. The majority of citation bursts occurred post-2000 and were attributed to four authors over a span of 10 years, indicating sustained interest in the subject in recent years and the enduring academic impact of these authors. Researchers can follow these influential and highly cited findings as references for future research directions. These influential authors provide valuable guidance and information on emerging trends for researchers, which may affect their research funding and collaboration.

### 2.5. Analysis of High-Contributing Journals

The visualization of journal publication data reveals that 377 journals have published articles on the effects of GAs on rice biological processes. A thermodynamic graph was constructed to demonstrate the distribution of this literature among the journals, with a minimum standard of eight articles per journal (shown in Figure 7). The color shade in the graph represents the distribution of journal articles. *Plant Physiology* had the highest number of papers (n = 113, 5.34%), followed by *Frontiers in Plant Science* (n = 94, 4.44%), *International Journal of Molecular Sciences* (n = 79, 3.73%) and *Plant and Cell Physiology* (n = 73, 3.45%). This information provides researchers with a comprehensive understanding of the publishing landscape in this field, aiding them in selecting appropriate journals for their work and ensuring that their research reaches a relevant audience. Additionally, Figure 8 presents the top 20 journals with the highest citation frequency for the papers on the effects of GAs on rice biological processes, enabling researchers to evaluate the significance of specific journals and prioritize them when citing or referencing previous research in their own work.

The double map overlay technique effectively illustrates the interdisciplinary distribution of journals, the trajectory of citation development, and the migration of research centers [15]. The map labels describe the subject areas covered by the journals, with the citing journals on the left and the cited journals on the right [16]. Different-colored lines visually represent citation paths, originating and terminating at the citation map. The breadth of these paths correlates with the frequency of z-score-scaled citations. Figure 9 categorizes the studies on the effects of GAs on rice biological processes into botany, ecology, molecular biology, immunology, and genetics. The authors also analyzed literature fields and visually clustered research articles using the VOSviewer software, dividing them into five major fields. In Figure 10, colored balls denote different clusters, with the yellow cluster of Ecology and Environmental Science having the highest number of articles (1695), particularly in the subfields of Plant Sciences, Agronomy, and Horticulture. Analyzing interdisciplinary and research hotspots in this field promotes collaboration among researchers from different backgrounds, facilitates the exchange of ideas and innovative methods, and thus accelerates the research on hot topics in this field.

### 2.6. Co-Cited References

It is important for researchers to be aware of the most attention-receiving and cited works. This can provide valuable insights and guide future research directions in their respective fields. Using CiteSpace, the co-citation of the literature related to the impact of GAs on biological processes in rice from 1 January 1985 to 29 February 2024 was analyzed. The size of the orbs in the visualization corresponds to the number of co-citations, with purple indicating early citations and yellow representing later citations. The connecting lines between the orbs show the co-citation relationships. The nodes marked in red have high centrality. The review entitled ‘An overview of gibberellin metabolism enzyme genes and their related mutants in rice’ by Tomoaki Sakamoto et al. in *Plant Physiology*, 2004, [17] was highly cited (n = 75), followed by the work ‘GIBBERELLIN INSENSITIVE DWARF1 encodes a soluble receptor for gibberellin’, by Miyako Ueguchi-Tanaka et al. [18], in *Nature*, 2005 (n = 63). CiteSpace facilitates the evaluation of network structure using modularity (Q-value) and clustering clarity (S-value), with values above 0.3 and 0.5, respectively, indicating significant clustering. The analysis revealed a Q-value of 0.8824 and an S-value of 0.9759, confirming a strong clustering structure in the network. Figure 11 showed that the co-cited literature has been grouped into 15 categories, such as plant height regulation, genome-wide identification, gene expression, upstream region, salicylic acid, deepwater rice, genetic manipulation, GA production, nuclear GA, endogenous GA, Della protein, functional identification, gras gene family, encoding GA, and promoter element. The clusters with low node counts and no research significance (#7, #14) were removed. Researchers can refer to this information to align their work with the established research content, thereby increasing the likelihood of their research being recognized and cited by peers and experts.

We utilized CiteSpace’s analytics to identify citation spikes, focusing on studies that have garnered considerable academic interest in the realm of the effects of GAs on biological processes in rice. Figure 12 presents an analysis of the top 20 references for notable impact and citation spikes. Among these, the work by Sakamoto T (2004) [17] exhibited the highest citation surge intensity at 32.4, with a surge timeframe of 2004–2009. The majority of the literature experienced citation surges between 1998 and 2013, underscoring the enduring importance of research on GAs’ effects on rice biological processes. Notably, Chen CJ (2020) [19] demonstrated a citation spike during the period of 2021–2024, suggesting that the bioinformatics approach has made a significant academic impact in recent years.

These pieces of information can help researchers identify new research areas and trends, and quickly shift their research focus to high-impact research areas. Analyzing co-cited references and identifying frequently referenced works can facilitate the identification of potential research collaborators with similar interests, enhancing the potential for collaborative projects. Partnering with authors whose work is frequently cited can enhance the visibility and impact of one’s research. Industry professionals can leverage insights on influential works and research trends to make informed strategic decisions, such as identifying research partners or investment opportunities in alignment with highly cited and influential research.

### 2.7. Keyword Analysis

Keyword co-occurrence cluster analyses were performed using the VOSviewer software program, with a minimum threshold of eight instances for keyword occurrences. Only the keywords meeting this criterion were included in the chart. From the initial 4196 keywords (de-merged and merged to 4124), 118 keywords were selected to create the visualized network as shown in Figure 13. In this visualization, circles and their corresponding labels create nodes, with the size of each circle reflecting keyword frequency and the thickness of connections indicating the strength of relationships. The color of each circle changes gradually to represent the average year of occurrence, with blue indicating earlier appearances and yellow indicating later ones. Notably, uniconazole, alpha-amylase, and submergence coercion emerged as earlier research topics, while Della, plant height, and abiotic stress are recent hotspots. This information provides valuable insights for researchers, highlighting the areas of increasing interest and potential for further exploration. The identification of keyword bursts, particularly those with significant increases in citations, reveals the areas of high academic interest, as depicted in Figure 14B. The data show that 20% of the keywords experienced a citation surge in 2019 and 2021, with 40% maintaining a high citation rate over the past 3 years. Most keyword surges occurred after 2016, indicating growing interest in the research area of GAs’ effects on rice biological processes. Notably, ‘plant height’ had the highest burst rate of 10.67, followed by ‘seed germination’ and ‘abiotic stress’ with burst rates of 7.86 and 7.7, respectively. These insights hold significant value for researchers in their respective fields.

Understanding the evolving trends and hotspots of research on the effects of gibberellins on rice biological processes is crucial for informing investment and collaboration decisions and ensuring alignment with current research interests. Figure 14A provides a temporal analysis of keyword frequency clustering within hotspots. The figure visually represents circles of varying sizes, where the cumulative diameter in each annual cycle corresponds to keyword frequency. Interconnections between the keywords denote co-occurrence, with purple denoting earlier appearing keywords, yellow representing more recent ones, and overlapping colors indicating the keywords present in all years. Rose nodes highlight the keywords with significant centrality, underscoring their importance as network hubs. The keywords within the same cluster are horizontally arranged, with the time of a keyword’s first appearance at the top and becoming more recent towards the right. This visualization aids in understanding keyword distribution across clusters, with higher numbers indicating more significant clusters. It also provides insight into the time span of the keywords in each cluster. The keywords were classified into 13 different groups: #0 transgenic rice plant, #1 expression profiling, #2 bulb dormancy, #3 gibberellin production, #4 floating rice, #5 wrky transcription factor, #6 cysteine endopeptidase gene, #7 secondary metabolism, #8 labdane-related diterpenoid metabolism, #9 deep-water rice, #10 chilling stress, #11 low temperature. These clusters encapsulate the primary themes and hotspots in the research area, reflecting their significance in scientific advancement. By presenting knowledge and research activities at a macro level, this visualization enables researchers to grasp overall trends.

## 3. Discussion

Through the above work, we obtained basic information on the impact of GAs on rice biological processes. Using bibliometric research methods, we started from the countries/regions, institutions, and journals where the articles were published, including the relevant high-impact researchers involved and the hot issues covered. As a result, a relatively comprehensive, objective, and scientific understanding of this field has been formed. Despite there being various areas that could be enhanced, this research has also produced notable findings.

### 3.1. Global Research Landscape

Our analysis of the effects of GAs on rice bioprocesses research from 1985 to 2024 reveals a consistent upward trend in annual publication volume (Figure 2), indicating that rice- and GA-related research is expanding. Based on the analysis of the highly cited literature (Figure 12), the GA- and rice-related research has gone through two phases: a slow development phase (1985–2000) and a rapid growth phase (2001–2024). Famous authors such as Murofushi N and Takahashi emerged in the first period. Most of their work focused on molecular cloning and the functional studies of GA metabolism-related genes, which laid a good foundation for later work. The later article by Tomoaki Sakamoto in *Plant Physiology* effectively deepened the understanding of rice (*Oryza sativa*) GA synthesis and metabolism by screening candidate genes for GA-metabolizing enzymes using the available rice gene database. From the 29 pre-selected genes, it was demonstrated that the enzymes catalyzing the early steps of the GA biosynthetic pathway (i.e., CPS, KS, KO, and KAO) are predominantly encoded by single genes, whereas the enzymes catalyzing the subsequent steps (i.e., GA20ox, GA3ox, and GA2ox) are encoded by gene families [17]. The importance of this work can be reflected in the fact that it is the most cited literature over a 40-year period. This groundbreaking article in the field is highly recommended for all researchers interested in the topic.

Upon further analysis at the research institution and country/region levels, it was revealed that the Chinese Academy of Sciences is the most influential and contributing institution, with China being identified as the most significant research country. Chinese Acad Sci is a renowned research institute and university in China, and is currently one of the world’s leading research organizations (Figure 4). The main monographs of this institution are on the effects of the upstream and downstream genes of GAs on phenotyping in rice. For example, the academics at this institution investigated the molecular mechanism behind the inverse relationship between plant height and tiller number, which led to gibberellin driving the green revolution in rice. Their analyses revealed that the rice tiller regulator MONOCULM 1 (MOC1) is protected from degradation by binding to the DELLA protein SLENDER RICE 1 (SLR1). GA triggered the degradation of SLR1, leading to stem elongation, and at the same time led to the degradation of MOC1, which reduced tiller number, further answering the mechanism of GA’s action in rice biological processes [20]. In short, highly cited scholars not only elevate the status of their research institutions but also elevate the status of their countries in specific fields or disciplines, keeping them at the forefront of research in that field. China leads in the number of published articles, with Japan and the United States following closely. This trend correlates with the rankings of research institutions and authors based on their publication output (Figure 3A). The main distribution region of rice also has a significant impact on the number of papers, with China’s research strength being widely recognized and its influence unquestionable. The Chinese Academy of Sciences and other research institutions in China play a significant role in advancing research in this field. By prioritizing research topics and actively seeking collaborations, interested researchers may be able to achieve impactful results.

### 3.2. The Effects of GAs on Rice Biological Processes

To investigate the impact of GA on rice biological processes, we conducted a statistical analysis of the highly cited literature and performed keyword analysis on the collected data. By integrating the characteristics and advancements in GA- and rice-related research, our examination was structured around three key areas: GA metabolism, GA and rice dwarfing, and current trends in GA research.

#### 3.2.1. Biosynthesis and Catabolism of GAs in Rice

Through the time trend analysis of keyword co-occurrence (Figure 14A), it was observed that research on the biosynthesis and metabolism-related gene expression of GAs were the predominant focus from 1985 to 2000. To the best of our knowledge, several scholars have conducted extensive research in this field, including Tomoaki Sakamoto et al.’s review of GA anabolic genes in rice [17], which is the most highly cited literature in this field. Tomoaki Sakamoto et al. summarized the previous work of Peter Hedden [4] and Stephen G. Thomas’s [21] studies of GA 20-oxidase, GA 3β-hydroxylase, and GA 2-oxidase genes, and screened for early synthetic genes such as CPS, KS, KO, and KAO, which paved the way for the later studies of the GA pathway. As the citation clustering analysis in Figure 12 shows, as early as 2000, Peter Hedden proposed that multifunctional enzymes reduce the types of enzymes required for the synthesis of endogenous GAs based on the genes identified at that time to be involved in the GA hormone metabolism pathway, but that enzymes encoded by multiple genes co-regulated by multiple genes increase the genetic complexity. The proposed discovery supported the subsequent 2007 study of the metabolism-regulating pathway by Ivo Rieu, which verified the importance of the activity of the GA biosynthetic enzyme, GA 20-oxidase (GA20ox), in maintaining endogenous GA concentrations in plants by reverse genetics in Arabidopsis thaliana. Thus, all the important genes in the biosynthesis and metabolism pathways of GAs have been identified. This discovery lays a foundation for future researchers to gain a deeper understanding of the fundamental impact of GAs on rice biology.

#### 3.2.2. Effect of GAs on the Height of Rice

The impact of GAs on plant height represents not only one of the earliest discoveries in this field but also remains a prominent research focus with a high frequency of investigation (Figure 14A). In order to solve the global food shortage problem, in the 1920s, many countries focused on ‘dwarfing genes’ to increase food production. Rice, a food crop, became one of the first targets of research, and in the 1950s, the first semi-dwarf 1 (Sd1) gene to control plant height was successfully identified, and excellent varieties of rice were successfully selected and bred to be resistant to falling, with yields up to 1000 kg per 667 m^2^ higher than those of taller varieties, which was known as the ‘first green revolution’ in the history of agriculture [22]. At the same time, the effect GAs have on plant height was discovered by analyzing the nature of 136 species; Ga screening was performed to determine the physiological effects of the plant growth regulation of GA1, GA3, GA4, and GA7, with GA3 found to be mostly exogenous, and with mostly GA1 and GA4 in the body of the plant [23]. GAs regulate rice plant height, and the dwarfing rice phenotype cannot help but draw attention to the fact that in order to study GA signal transduction, many scholars have made previous efforts with plant height as an entry point. A cluster analysis of citations (Figure 13) reveals that Wolfgang Spielmeyer localized the semi-dwarfing gene SD-1 to the GA 20-oxidase gene by the gene mapping localization and GC-MS detection of GA content in elongated stems [24]. This was reaffirmed by A. Sasaki, and published in Nature in the same year [25]. In addition to the dwarfing phenotype, Akira Ikeda investigated a rice slender mutant (slr1-1) with a constitutive GA phenotype, and concluded that the product of the SLR1 gene is an intermediate in the GA signal transduction pathway [26]. This was followed by Hironori Itoh in 2002, who analyzed the phenotype of the transformants and the subcellular localization of GFP in vivo by constructing transgenic rice expressing a fusion protein consisting of SLR1 and a green fluorescent protein (SLR1-GFP), and obtained a further explanation that GA signaling is regulated by the appearance or disappearance of the nuclear SLR1 protein, which is controlled by upstream GA signaling control, as was further explained [27]. Akie Sasaki confirmed the identity of SLR1 as a blocker of GA signaling by isolating and characterizing the GA-insensitive dwarf mutant gid2 in rice, and put forward the theory that SLR1 is degraded through GA-dependent phosphorylation by the SCF (GID2)-proteasome pathway [28].

On the basis of the analysis of the keywords, Della and GID1, the two most important components of the GA signal transduction pathway, have only slowly begun to surface. The analysis of Della’s expression profile emerged as a popular research focus during that period (Figure 14A). In 2005, Miyako Ueguchi-Tanaka found a series of results such as the binding of GID1 to the rice DELLA protein SLR1 in a GA-dependent manner in yeast cells, demonstrating for the first time that GID1 is a soluble receptor that mediates GA signaling in rice [18]. Subsequently, Miki Nakajima demonstrated this conclusion in Arabidopsis thaliana [29]. In 2007, Miyako Ueguchi-Tanaka proposed a molecular model for the interaction between GA, GID1, and SLR1 [30], which became an article of extraordinary impact then and now, with a citation burst strength of 30.99 during the period of 2006–2010, ranking second in the last 40 years. The crystal structure of the GID1-GID1-DELLA complex is the same as that of the GA-GID1-GID1 and GA-GID1-DELLA complexes. The resolution of the crystal structure ‘closes the lid’ on GA signaling [31]. Through detailed research on the della protein, it also has been discovered that there is a significant crosstalk between GAs and various hormones involving this protein. The establishment of this framework enhances our comprehension of the GA signaling pathway in rice, making it essential knowledge for researchers in this particular field.

#### 3.2.3. Research Trends over Recent Years

In recent years, in addition to the keyword ‘plant height’, research on seed dormancy and germination, stress resistance, root growth, heading, and other related topics have gained increasing attention in this field (Figure 14A). A 2016 review summarized a complex molecular network of phytohormones that play key roles in regulating seed dormancy and germination, with AP2 domain-containing transcription factors playing key roles, and the ABA/GA balance constituting a central node; the balance of ABA/GA constitutes the central node of this crosstalk of different hormones [32]. This research direction became a hotspot during the period of 2019–2022, with a burst rate of 7.86. In 2019, Qian Li and Baolan Wang et al. focused even more on endogenous GA activity and seed sensitivity to salt stress [33] and increased NH4+ toxicity sensitivity [34]. Abiotic stress is a hot research topic after plant height and seed germination, with a burst rate of 7.7 between 2021 and 2024, and this pattern of combining hot research topics into a publication may become a research trend in this field. The GA signaling pathway repressor DELLA protein is a central regulator that mediates the crosstalk of various phytohormones [35], so the effects of GAs on rice biological processes are usually accompanied by crosstalk with other hormones. Based on this idea, in 2020, Huaqin and his colleagues focused on the regulatory effects of GAs and ethylene on primary root growth, and proposed that GA acts downstream of the ethylene signaling pathway to control cell proliferation in the rice primary root meristem [36]. The same year, Jintao Li et al. proposed that GAs regulate local root elongation by regulating OsYUCCA6 and PIN expression, and that GAs regulate local root elongation through regulating OsYUCCA6 and PIN expression by analyzing the root elongation of rice seedlings and DR5-GUS activity. In addition, the authors proposed that PIN expression regulates local growth hormone biosynthesis and polar growth hormone transport, providing new insights into how GAs affect rice root elongation [37]. Spike architecture is a key determinant of grain yield, and Su Su et al. proposed that SLR1 physically interacts with the phloem identity class I KNOTTED1-LIKE HOMEOBOX (KNOX) protein OSH1 to repress the OSH1-mediated activation of downstream genes related to spike development, providing a mechanistic link to the effects of GAs on spike morphogenesis in rice [38]. Furthermore, Kun Wu analyzed the mechanism of rice tillering and the downstream protein of the GA signaling pathway, NGR5, from an epigenetic perspective: the increase in the activity of NGR5 decoupled tillering from nitrogen regulation, which ensured rice yield under the low-nitrogen fertilizer application level [39]. A thorough understanding of current research trends can empower researchers to conduct more meaningful and valuable research. Recent research trends have shifted from basic theoretical discoveries to innovative findings with potential applications in agricultural guidance, as evidenced by an analysis of the popular keywords.

## 4. Materials and Methods

### 4.1. Data Source

The Web of Science Core Collection database (WoSCC, Clarivate Analytics, Philadelphia, PA, USA) is an authoritative and specialized citation database renowned for its robust indexing capabilities. It has been extensively utilized in previous bibliometric studies due to its comprehensive content, which includes not only basic information such as title, author, institution, country/region, and author keywords, but also detailed reference information [10]. In this study, we conducted a search for publications related to gibberellins and rice in the WoSCC of Science Citation Index Expanded (SCIE), spanning from 1 January 1985 to 29 February 2024.

### 4.2. Data Search Strategy

The authors conducted an extensive online search within a single day to collect all the potentially relevant publications related to gibberellins in rice. The search query employed was [TS = (Gibberellins OR Gibberellin) AND TS = (rice OR ‘*Oryza sativa*’)], concentrating on studies published between 1 January 1985 and 29 February 2024. Only English-language articles and reviews were included, with specific exclusion criteria detailed in Figure 1. A total of 2264 documents were retrieved, out of which 2118 articles and reviews were chosen for the data analysis after a thorough manual screening of titles, abstracts, and full texts by the researcher.

### 4.3. Data Extraction and Collection

During the data extraction and collection process, a total of 2118 documents were gathered. These documents were identified as fully documented and cited references in plain text or tab-delimited (win, UTF-8) format for analysis using bibliometric tools. The analysis involved studying various bibliometric indicators, including annual publication and citation counts, country/region, institution, author, funding agency, journal, keywords, and research area using Microsoft Excel 2021. To overcome some of the limitations of the WOS database, the study incorporated specific data within countries. Journal Impact Factor (JIF) and subject category quartile rankings were sourced from the Journal Citation Reports (Clarivate, 2023), with the journals categorized into four quartiles based on JIF values. Additional bibliometric data, such as total citation time, average number of citations, H-index, etc., were obtained from the Citation Report function of WoSCC.

### 4.4. Bibliometric Analysis

In order to conduct a more comprehensive analysis of the data, three bibliometric software packages were utilized: VOSviewer 1.6.18, Citespace 6.3.R1, and Pajek 64 5.16. Additionally, we employed the chorddiag R language package and two online websites for visualizing and analyzing various aspects including country/region, institution, authors, journals, fields, co-citations, and keywords. These tools were employed to draw relevant visual maps in order to analyze the current status of the research, research hotspots, and keywords, as well as to identify trends in the research.

VOSviewer, a Java-based software developed by van Eck and Waltman for bibliometric mapping and cluster analysis, was used in this study [40]. VOSviewer 1.6.18, combined with the chorddiag R language package, was utilized for the country/region relationship analysis and visualization. Furthermore, VOSviewer 1.6.18 and Pajek 64 5.16 were employed for examining the co-occurrence of country/region, institution, author, domain, and keywords. The cooperation between the countries was displayed using a website (https://bibliometric.com), while the frequency trends of keywords were visualized using an additional website (https://www.citexs.com).

In this research, the study utilized CiteSpace, a software tool created by Professor Chaomei Chen, to analyze and visualize bibliometric networks [41]. The software version employed was Citespace 6.3.R1 (Chaomei Chen, China). The visualization maps generated included countries/regions, institutions, authors, journals, literature co-citations, and keywords. The top 10 citation burst intensities for the countries, institutions, and authors were mapped, while the top 20 citation burst intensities for the journals and the literature were also visualized. The literature co-citation clustering analysis map utilized CiteSpace parameters with time slices set at (1985–2024), years per slice (1), and selection criterion (k = 3). Likewise, the timeline analysis for keyword clustering timeline analysis employed CiteSpace parameters set at time slice (1985–2024), year per slice (1), and selection criteria (k = 10).

### 4.5. Statistical Analysis

Data analysis focusing on description, graphing, and curve fitting were conducted using Microsoft Excel 2021, R software (v 4.3.2), and the GraphPad Prism software (v 9.0). The quantity of publications and citations per year was computed in Microsoft Excel, and curves were adjusted using various functions such as exponential, logistic, logarithmic, and polynomial. The best-fitting model was chosen based on the strength of the correlation coefficient (R2). The rate of publication increase over time was deduced by employing a specific formula: Growth rate = [(number of publications in the previous year ÷ number of publications in the first year)1/(last year − first year) − 1] × 100 [42]. The Pearson correlation coefficient test was employed to evaluate the relationship between publications and citations, with a *p*-value of <0.05 indicating a significant Pearson correlation.

## 5. Conclusions

This study conducted a comprehensive analysis of countries, institutions, authors, journals, keywords, and other aspects, thereby revealing the characteristics of gibberellin-related research in rice. The increasing trend of published articles underscores the importance of gibberellin in ongoing rice research. China and the United States emerged as prominent contributors, leading in the number of articles published and engaging in both domestic and international collaborations. The Chinese Academy of Sciences stands out as an active contributor to the field, with its professors and researchers making significant academic contributions. Research findings from Nagoya University hold substantial influence and are widely cited in the academic community, driving progress and advancements in the field. Author Lee In-Jung is acknowledged for the significant impact, while Matsuoka M., is acclaimed for academic excellence, garnering attention from researchers. With a substantial number of related articles published, Plant Physiology is becoming a key journal for gibberellin research in rice.

It has been demonstrated that the predominant focus of rice research has historically been on the synthesis and metabolism of gibberellin and the identification of core proteins in gibberellin signaling pathways. The dominant research trend in this field has gradually shifted from the plant height phenotypes affected by gibberellin in the past to other phenotypes such as seed dormancy and germination, stress resistance, and root growth.

These findings are crucial for identifying upcoming trends and frontiers in gibberellin research in rice, thereby enhancing the ability of the scientific community to stay updated with the current research landscape. It is important for researchers to stay informed about these evolving trends and leverage existing knowledge to propel the field forward. However, there are still gaps in the research in this field. For instance, the intricate interrelationships between the proteins in the rice gibberellin signaling pathway have yet to be elucidated, and the mechanisms underlying gibberellin’s impact on rice phenotypic development still need further study. Consequently, future research should prioritize fostering domestic and international collaboration to deepen the understanding of gibberellin-related research in rice. The integration of cutting-edge technologies such as epigenetics and functional genomics will support researchers in uncovering the biological mechanisms associated with gibberellins in rice. By employing collaborative, interdisciplinary, and methodical strategies, researchers can contribute to the development of rice cultivation in agriculture. This will improve human living conditions and overall well-being by enhancing the comprehension of gibberellin’s involvement in rice’s intricate biological processes.

## Figures and Tables

**Figure 1 plants-13-01548-f001:**
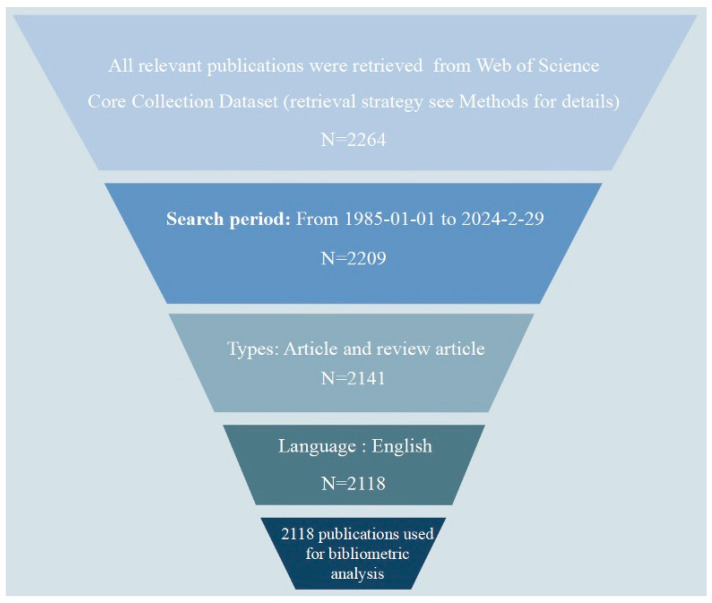
Schematic representation of the literature search and selection process.

**Figure 2 plants-13-01548-f002:**
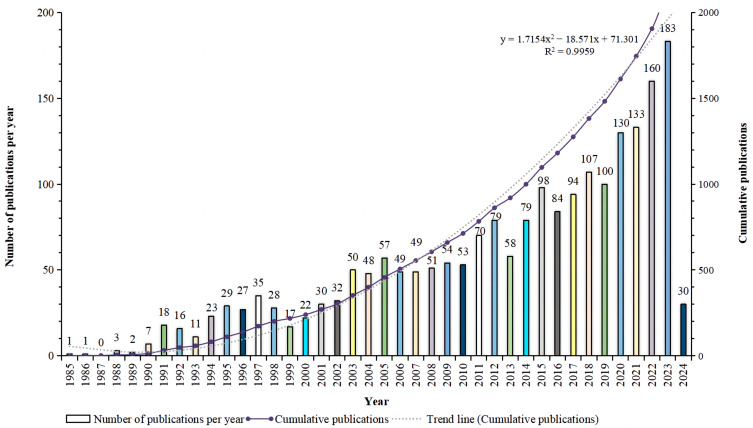
Trend analysis of research related to the effects of GAs on rice biological processes from 1985 to 2024.

**Figure 3 plants-13-01548-f003:**
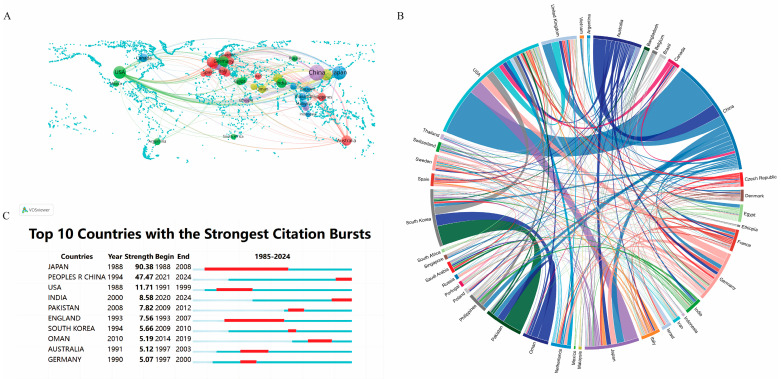
(**A**) Global distribution of the effects of GAs on rice biological processes research; (**B**) chord diagrams illustrating international collaborations; (**C**) research output on the effects of GAs on rice biological processes from the top 10 countries (the blue bar represent the period of citations and the red bar represent the periods of citation bursts).

**Figure 4 plants-13-01548-f004:**
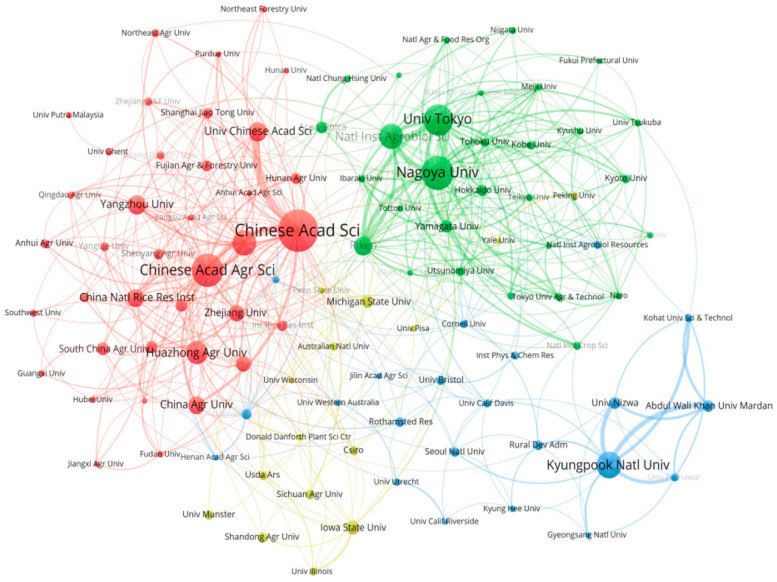
Schematic diagram of cluster analysis of institutional cooperation.

**Figure 5 plants-13-01548-f005:**
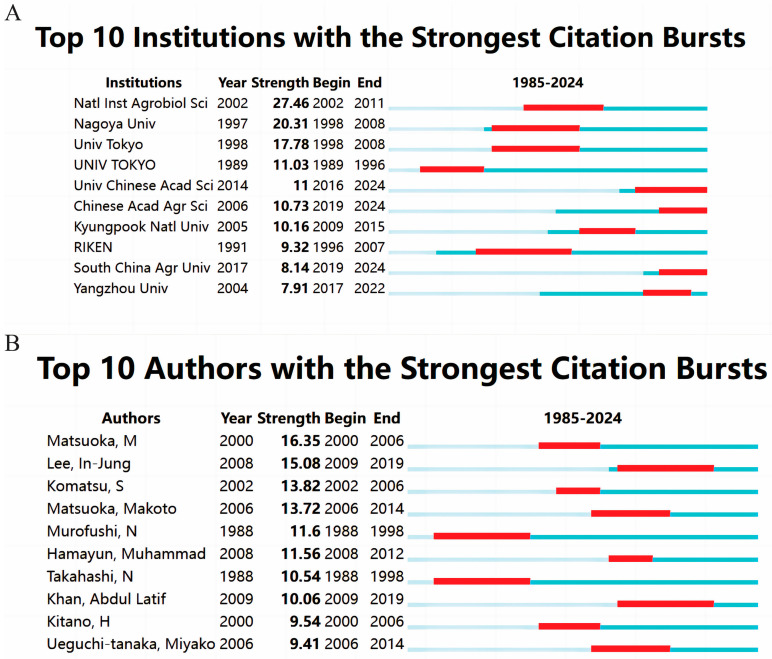
(**A**) Top 10 institutions with citation bursts; (**B**) top 10 authors with significant citation bursts in publications on the effects of GAs on rice biological processes (the blue bar represents the period of citations and the red bar represents the periods of citation bursts).

**Figure 6 plants-13-01548-f006:**
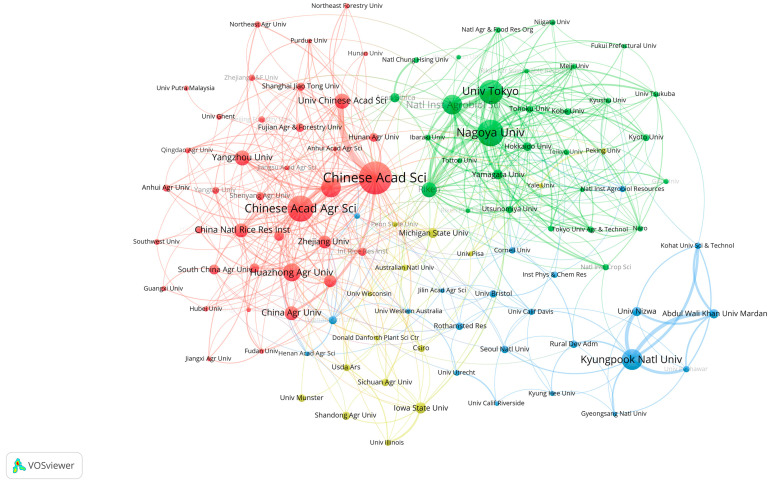
Author co-occurrence analysis chart map.

**Figure 7 plants-13-01548-f007:**
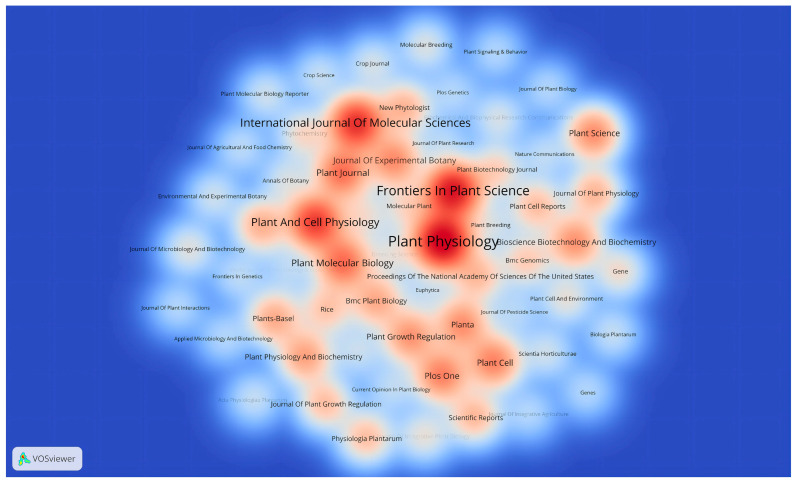
Density visualization map of journal citations.

**Figure 8 plants-13-01548-f008:**
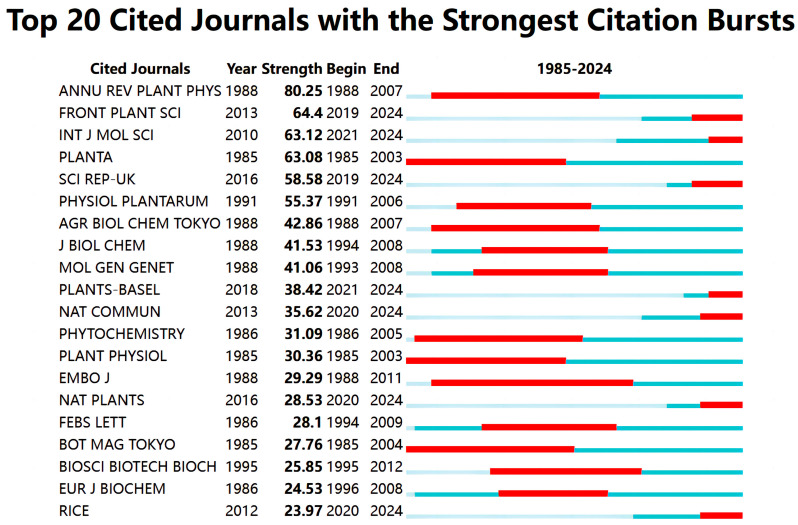
Top 20 journals with significant citation bursts (the blue bar represents the period of citations and the red bar represents the periods of citation bursts).

**Figure 9 plants-13-01548-f009:**
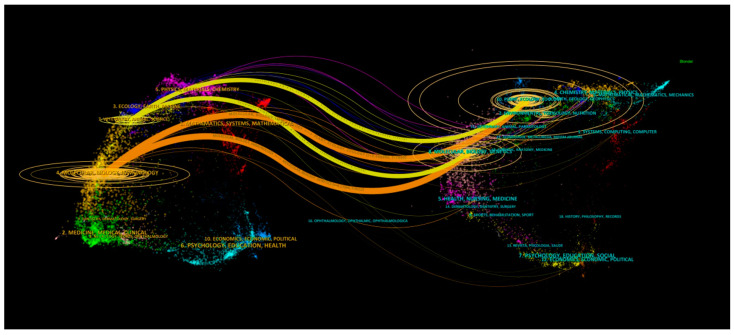
Dual-map overlay of journals.

**Figure 10 plants-13-01548-f010:**
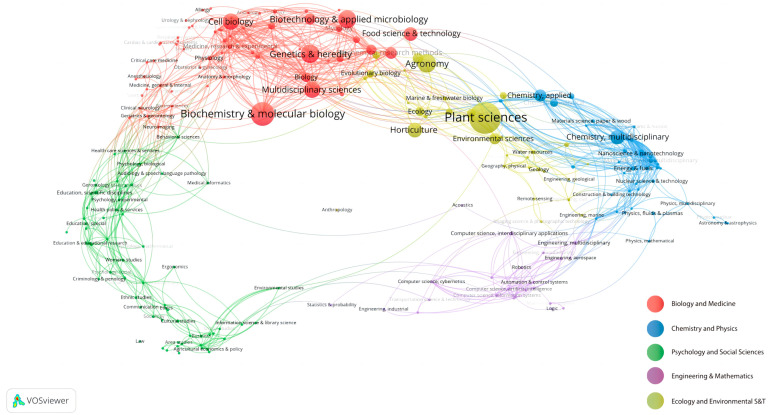
Domain analysis diagram for research on the effects of GAs on rice biological processes.

**Figure 11 plants-13-01548-f011:**
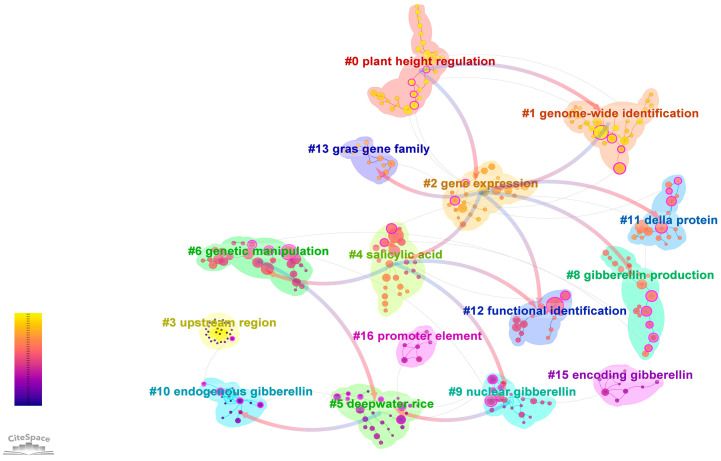
Co-citation analysis chart for research on the effects of GAs on rice biological processes.

**Figure 12 plants-13-01548-f012:**
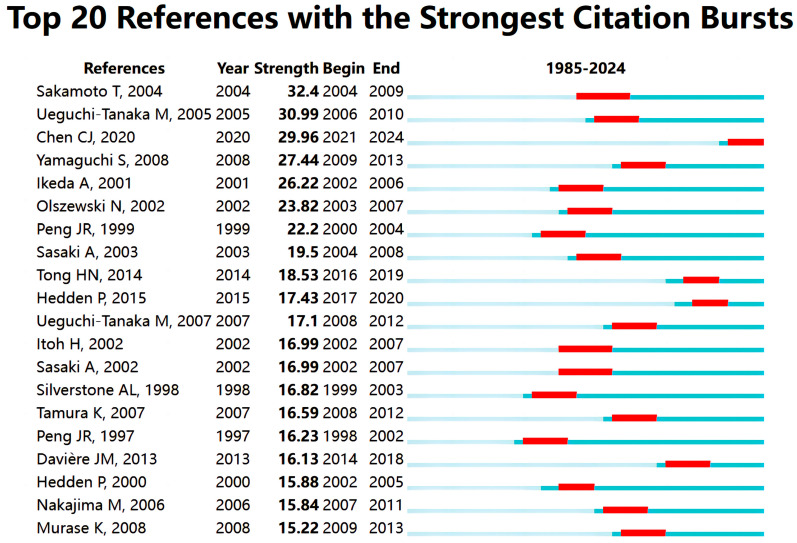
Top 20 references with the highest citation bursts (the blue bar represent the period of citations and the red bar represent the periods of citation bursts).

**Figure 13 plants-13-01548-f013:**
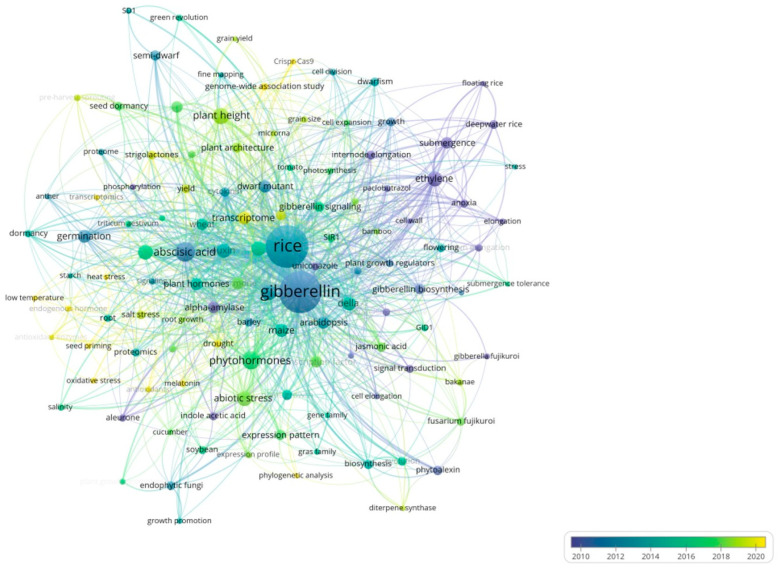
Keyword intensity visualization timing overlay.

**Figure 14 plants-13-01548-f014:**
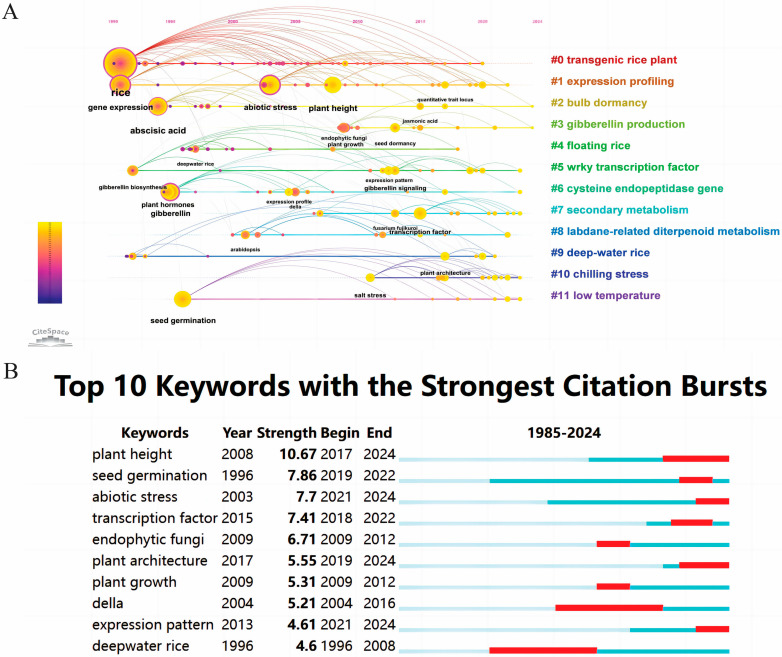
(**A**) Temporal trends in keyword co-occurrence; (**B**) top 10 keywords with significant citation bursts (the blue bar represent the period of citations and the red bar represent the periods of citation bursts).

## Data Availability

Data are contained within the article.

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
