# Peer review of "Research Overview and Trends of the Effects of Gibberellins (GAs) on Rice Biological Processes: A Bibliometric Analysis"

_plants, 2024, doi:10.3390/plants13111548_

Round 1

Reviewer 1 Report

Comments and Suggestions for Authors

The manuscript provides some valuable information on the overview and trends of GA on rice biological processes.

I would like to make the following comments.

The Abstract needs major revision. Please state clearly the objectives of your research study, the reasons that you undertook this study and the innovations for the other researchers. What was your justification and main goal for writing this paper?

I find that the manuscript does not have a clear objective and this shows in the results and discussion since some of the discussion lacks focus.

The increasing trend in the annual publication rate is seen in many areas of research and part of it has to do with the fact that a lot of journals exist today, and some of them are open access which is a more recent model of publishing research manuscripts. In the 1980s and 1990s, only few journals were available and scientists did not have as many opportunities to publish their research.

Thus, the overall increasing rate of publication may not necessarily translate to what the authors describe.

I find that some of the Figures are interesting and nicely convey specific information.

I suggest that the authors revise their manuscript based on my comments and also discuss how the results of this study would benefit the other scientists. The phrase “Keeping pace with these technological advancements will be crucial for addressing current issues in the field” is very general. Please revise.

Comments on the Quality of English Language

Minor editing of English language is required

Author Response

The manuscript provides some valuable information on the overview and trends of GA on rice biological processes.

I would like to make the following comments.

The Abstract needs major revision. Please state clearly the objectives of your research study, the reasons that you undertook this study and the innovations for the other researchers. What was your justification and main goal for writing this paper?

Response: We appreciate the reviewer's suggestion. We made significant changes to the abstract section, adding the purpose of conducting the study, adding a statement of the empirical experience, and clarifying the significance of the research in lines 16-29.

I find that the manuscript does not have a clear objective and this shows in the results and discussion since some of the discussion lacks focus.

Response: We appreciate the reviewer's suggestion. We added a description of the research object in the introduction section (lines 88-91) and reiterated it in the discussion section (lines 361-367) to provide clarity on the focus of the article and enhance the specificity of the discussion.

The increasing trend in the annual publication rate is seen in many areas of research and part of it has to do with the fact that a lot of journals exist today, and some of them are open access which is a more recent model of publishing research manuscripts. In the 1980s and 1990s, only few journals were available and scientists did not have as many opportunities to publish their research.

Thus, the overall increasing rate of publication may not necessarily translate to what the authors describe.

Response: Thank you for your valuable comments and we recognize that our previous interpretation of the phenomenon was not comprehensive enough. After considering the impact of open access journals on the number of publications, we believe that (1) the ease of dissemination and citation of the journals themselves, and (2) the fact that the relevant research has become topical again, are two or more reasons that have contributed to the increase in the number of research papers in this field, and we have added a related comment in line 105-110.

I find that some of the Figures are interesting and nicely convey specific information.

I suggest that the authors revise their manuscript based on my comments and also discuss how the results of this study would benefit the other scientists. The phrase “Keeping pace with these technological advancements will be crucial for addressing current issues in the field” is very general. Please revise.

Response: The reviewer’s suggestion has been adopted. The impact of advanced technologies on this field of research is further explored, with specific opinions provided on the future development of the field in lines 625-633.

Reviewer 2 Report

Comments and Suggestions for Authors

The abstract effectively communicates the scope and findings of the study, providing a comprehensive overview of the impact of GA on rice biological processes based on a bibliometric analysis. However, it could be enhanced by including more specific details about the research gaps and potential future directions in this field.

1. The abstract focuses mainly on the analysis of existing literature and does not mention any new empirical findings or experimental results.

2. While this study discusses trends in GA research in rice, it does not provide specific details about the biological processes affected by GA or the specific research gaps identified.

3. Line 24-25 could cite Inbred varieties outperformed hybrid rice varieties under dense planting with reducing nitrogen. Scientific Reports10(1), 8769.

4.Line 63-64 could cite Biochemical response of okra (Abelmoschus esculentus L.) to selenium (Se) under drought stress. Sustainability 15, 5694.

Author Response

The abstract effectively communicates the scope and findings of the study, providing a comprehensive overview of the impact of GA on rice biological processes based on a bibliometric analysis. However, it could be enhanced by including more specific details about the research gaps and potential future directions in this field.

  1. The abstract focuses mainly on the analysis of existing literature and does not mention any new empirical findings or experimental results.

Response: We appreciate the reviewer's suggestion. Based on the data obtained from our analysis, we present an overview of the current research landscape in this field, highlighting key findings and insights. Additionally, we identify research gaps and propose potential future directions for advancing the field. in the lines 21-29.

  1. While this study discusses trends in GA research in rice, it does not provide specific details about the biological processes affected by GA or the specific research gaps identified.

Response: We appreciate the reviewer's suggestion. We have made modifications to section 3.2 of our study, focusing on three main areas: gibberellin metabolism, gibberellin and rice dwarfing, and the latest developments in gibberellin research. Furthermore, We have enriched the discussion of the latest findings on the effects of GA on biological processes in rice by  adding relative references to this paragraph in lines 504-507,513-518.

  1. Line 24-25 could cite Inbred varieties outperformed hybrid rice varieties under dense planting with reducing nitrogen. Scientific Reports10(1), 8769.

Response: We appreciate the reviewer's suggestion. We revised the relative contents, and added relative references to this paragraph in Lines 33-35.

4.Line 63-64 could cite Biochemical response of okra (Abelmoschus esculentus L.) to selenium (Se) under drought stress. Sustainability 15, 5694.

Response: We appreciate the reviewer's suggestion. We revised the relative contents, and added relative references to this paragraph in Lines 76-79.

Round 2

Reviewer 1 Report

Comments and Suggestions for Authors

Authors have responded to my comments and the manuscript has been significantly improved. It can now be published in Plants